# Prevalence of child malnutrition and household socioeconomic deprivation: A case study of marginalized district in Punjab, Pakistan

**Muhammad Shahid[1], Farooq Ahmed[2], Waqar Ameer[3]\*, Jing Guo[4], Saqlain Raza[5], Saireen Fatima[6], Madeeha Gohar Qureshi[7]**

1 World Health Organization (WHO), Balochistan, Pakistan, 2 Department of Anthropology, Quaid-i-Azam University, Islamabad, Pakistan, 3 Department of Economics, Shandong Technology and Business University, Yantai, China, 4 Department of Health Policy and Management, School of Public Health, Peking University, Beijing, China, 5 Respiratory Care Department, College of Applied Medical Sciences in Jubail, Imam Abdulrahman bin Faisal University, Jubail, Saudi Arabia, 6 Fazaia Medical College, Air University, Islamabad, Pakistan, 7 Department of Economics, Pakistan Institute of Development Economics, Islamabad, Pakistan

\* waqar.ameer@yahoo.com

**Data Availability Statement:** All the relevant data and Supporting File information are available within the article. The Ethics Statement is given in the methodology part of the paper.

## Abstract

Better socioeconomic status and well-being in households decrease malnutrition and health risks in children. The objective of the present study is to assess the current nutritional status of pre-school children and to correlate the prevalence of malnutrition with Household Deprivation Status (HDS) in one of the deprived districts of the Punjab province in Pakistan. Using primary data collected from 384 households through a proportional purposive random sampling technique, this study calculates the z-scores of weight-for-age (WAZ), weight-for-height (WHZ), and height-for-age (HAZ). The study has used a cut-off point which is -2 standard deviations below the median of the WHO/NCHS reference population for each anthropometric indicator. The results indicate that the underweight, stunting, and wasting prevalence rates are 46.1%, 34.83%, and 15.49% respectively in district Rahimyar Khan. Also, the expected tendency of malnutrition is worst for HDS-1 and HDS-2 which are the most deprived segments of the population. As the household shifts from HDS-1 to HDS-2 and further to HDS-3, the rates of stunting (HAZ) and underweight (WAZ) decreases but wasting (WHZ) does not. The study concludes that the high prevalence of malnutrition in the district is correlated with overall socio-economic deprivation.

## Introduction

Household poverty and child undernutrition reinforce each other [1]. According to Srinivasan and Mohanty [2], Household Deprivation Status (HDS) has a substantial impact on child nutrition health status. HDS leads children to malnutrition, and resultantly, drops the capacity to work in adulthood which leads to extreme poverty. The same circle repeats for upcoming future generations by leading them to destitute poverty [2]. Children with better nutrition

**Funding:** The funding of this study was supported by the National Social Science Foundation of China (Number: 21BJY113).

**Competing interests:** The authors have declared that no competing interests exist.

have advanced development scores on social, cognitive, and emotional scales than those who are malnourished [3]. More than one-third of the population in Pakistan are living their lives below the poverty line [4].

Half of the Pakistani population have no access to proper sanitation facilities [5] and about 42 percent have no access to proper formal education [6]. Hence, one factor strengthens or substitutes the other in making the poor population the most vulnerable group that brings adverse health and nutritious effects [7]. The poor are the hardest hit mass mainly due to their limited economic resources and their low capacity of bearing the healthcare cost. Investment in health and nutrition or providing social protection potentially safeguard people against hostile socioeconomic circumstances [6, 7]. During the East-Asian economic crisis of 1997–98, a study illustrated that the economic crisis led to a substantial reduction in health service utilization by 25% in Indonesia while health care utilization increased in Thailand in the same period because of health insurance programs [8]. To improve household social and economic status, they need more resources to provide to their children–including healthy food to boost nutrition as well as proper medication in case of any disease [8].

In the district Rahimyar Khan, more than 78 percent of people live in rural areas [9] where the prevalence of malnutrition and child mortalities is the third-highest among the thirty-six (36) districts in the province of the Punjab [10, 11]. In this district, a large number of families in the lowest wealth index quantile along with low literacy and employment have to face difficulties to meet their ends [10]. Most of the existing literature which discusses the causes of malnutrition emphasizes that poverty or poor socioeconomic status is the main cause behind malnutrition prevalence. The existing literature in Pakistan only guides us that wealth status or poor socio-economic status is one of the significant determinants of malnutrition. But there is a need for further study which can examine the in-depth association between malnutrition and poor socio-economic status (household deprivation). Our study will cover this research gap by assuming the link between child nutritional status with the different household deprivation levels by arguing whether household deprivation status is contributing to child malnutrition in one of the marginalized district of highly populated province or not. More clearly, the study examines the current nutrition status in pre-school children of rural Rahimyar Khan. The study also correlates the child's nutritional status with the different household deprivation levels besides assessing the magnitude of malnutrition rates.

## Materials and methods

### Study area, sampling and data collection

A self-administered questionnaire was designed to collect primary, cross-sectional data from households living in the rural Rahimyar Khan. According to the census of Pakistan which was held in 2017, Rahimyar Khan District consists of 4 sub-district units or tehsils with a total population of 4,814,006 and total households are 701,520. The main source of economic activity of Rahimyar Khan is agriculture.

To avoid sampling bias in different ways, we determined the standard sample of 384 households assuming a 95% confidence level at a 5% level of significance. However, the design of the sample was established based on the probability proportional to size (PPS) in all 4 tehsils. The sampling frame consisted of all rural households in the district. The first stage was a stratified random sampling of rural clusters (villages; also called Union Councils (UCs)) in every tehsil or subdivision. In the second stage, a fixed number of households were selected and interviewed by following purposive sampling. In other words, our sample was representative at the tehsil level. Table 1 shows the distribution of sample size from Tehsils to Union Councils.

The rural households for the survey were chosen randomly through the record which was found in the lady health workers' register. The inclusion criteria of households were pre-school

**Table 1. Distribution of sample size from tehsils to union councils.**

| District | Tehsil | Union council No | Name of union council | ~Household |
|---|---|---|---|---|
| Rahimyar | Khanpur | UC-1 | Bagho Bahar | 26 |
| Khan | | UC-2 | Azeem Shah | 34 |
| | | UC-3 | Kotla Pathan | 36 |
| | Rahimyar | UC-7 | Bahishti | 34 |
| | Khan | UC-8 | Sonak | 46 |
| | | UC-9 | Chak No. 84/P | 35 |
| | Liaquatpur | UC-4 | Ghooka | 25 |
| | | UC-5 | Shadani | 26 |
| | | UC-6 | Trinda Gurgaij | 30 |
| | Sadiqabad | UC-10 | Kot Sanger Khan | 33 |
| | | UC-11 | Muhammad Pur | 32 |
| | | UC-12 | Roshan Bhet | 27 |
| Total | 4 | 12 | | N = 384 |

children considered for the household sample. The lady health workers were trained enough on how to take anthropometric measures before assigning the task. The rest of the information was collected by the principal investigator regarding socioeconomic status.

The majority of the mothers (87%) belonged to the 18–25 years age group. Around 58% of households' income was less than 50,000 Pakistani Rupees (US $320) per annum, while 26% of households' income was less than 1,00,000 Pakistani Rupees (US $640) per annum. It shows that about 93% of the selected households belonged to the HDS-1 (3%) and HDS-2 (90%) category which is the most deprived segment of the society. A total of 517 children was assessed. Out of 517 children, 286 (56%) were males and 231 (44%) were females. After completion of the data collection and data cleaning, the samples were taken for final analysis.

During the survey, if more than one family were found at a single boundary, we considered them nuclear if each family prepare their food independently. Data was gathered during three months in the study area from November 2017 to January 2018. S1 Table explains the distribution of samples in detail.

## Ethics statement

The graduate research management council (GRMC) approved the survey protocols in the sixth meeting on 16[th] June 2016 which was organized at the Pakistan Institute of Development Economics (PIDE). The GMRC at PIDE works as an institutional review board (IRB) in any research organization. The department of health economics at PIDE and department of health District Rahimyar Khan also approved survey protocols and tools. In tools, we used MUAC tape, weight machine, and height measurements tape for collecting data on height, weight, age of children and mothers. Additionally, we explained all the study details to the children's mothers. After briefing the objectives of the survey in the local languages (Punjabi or Saraiki), only verbal consent was obtained from the mothers as the majority of the mothers (74%) had no formal education and they also showed some reluctance due to their cultural bounds.

## Measures

This study has two objectives. First, it measures the child nutritional status of pre-school children; second, it investigates the relationship between child nutritional status with household

deprivation levels/quantiles to check the magnitude of malnutrition in different HDS categories.

**Undernutrition assessment.** After gathering the data on weight and height, anthropometric indicators (stunting, wasting, and underweight) were constructed based on comparison with a "healthy" reference population provided by WHO [12]) and National Center for Health Statistics (NCHS) [12]. Cut-off point- is two (2) standard deviation (SD) under the median of that reference population of WHO/NCHS was used for each of the anthropometric indicators for the measurement of child nutritional status [12]. The null hypothesis assumes that the child under study is not malnourished. The objectives of cut-off points taken into consideration were to classify the child according to nutrition status. To classify a child as moderately stunted, wasted, and under-weight, deviation from reference population z-scores $<$-2 SD were used, and further deviation of the z-scores $<$-3 SD place the child in the category of severe undernutrition.

Additionally, the study constructed a CIAF index (Composite Index of Anthropometric Failure) to see the overall malnutrition prevalence in children. According to CIAF classification, children are divided into seven groups which are as follows:

A: No Failure, B: Stunted only, C: Wasting only, D: Underweight only, E: Stunted and underweight, F: wasting and underweight, and G: stunting, wasting, and underweight. The total measure of child malnutrition prevalence was calculated by combinations of all groups except group A.

To construct HDS, we make use of the index provided by Srinivasan and Mohanty which considers socio-economic possession of the household [2, 13]. The measurement of HDS depends on three dimensions of household deprivation: basic economic possessions, basic amenities of life, and basic communication with the world. In HDS, six binary variables were used: 1) Household is constructed with mud or brick; 2) Household has some land or not; 3) Electricity is available in the house or not; 4) Drinking facility available or not in the residence; 5) Any member of the household is literate or not; 6) Keeping T.V, radio or newspaper in the house or not. In all three dimensions in HDS, adding these six variables shows total scores and the range of the scores from 0 to 6. Those who have none of any items from the six possessions or just have 1 or 2 items, were included in HDS-1 and were categorized as "moderate deprivation" (MD). Just above the deprivation (JAD) indicates those who have possession of any 3 items were categorized as HDS-2. HDS-3 includes those who had 4 or 6 items, it indicates "well above the deprivation (WAD)". The HDS does not directly measure household economic conditions such as total expenditure, per-capita income, or living standard index rather it measures households above the three dimensions that are deprived.

**Statistical analysis.** After the construction of stunting, wasting, underweight and CIAF variables, the descriptive statistics were taken to measure the prevalence of malnutrition in children. The cross-tabulations and pivot tables were computed by using STATA 14 software and Excel 13. Stunting, wasting, and underweight variables were constructed as a binary where "1" represents if the respective child is stunted/wasted/underweight, and "0" otherwise. Similarly, CIAF was also constructed as a binary variable where "1" represents if the respective child is malnourished, and "0" otherwise. Moreover, statistical interactions were examined to ascertain whether the relationship between WAZ, WHZ, and HAZ is moderated by the age of the child. Association between different anthropometric indicators was assessed through a two-way scatter plot. Before performing statistical analysis, the data was cleaned and ambiguities were removed accordingly. Z-scores which were outside the WHO flags were skipped from the dataset. Out of a total of 517 Under-Five children, 316 fulfilled the inclusion criteria and were included in the study. Descriptive statistics, chi-square test, and visualization were applied to examine the

**Table 2. Prevalence of underweight, stunting and wasting by sex of the child.**

| | Prevalence of underweight, stunting and wasting by sex of the child | | | | | |
|---|---|---|---|---|---|---|
| | Underweight (n = 269) | | Stunting (n = 267) | | Wasting (n = 71) | |
| | Moderate | Severe | Moderate | Severe | Moderate | Severe |
| Female | 33 (12.27%) | 58 (21.56%) | 50 (18.73%) | 48 (17.98%) | 8 (11.27%) | 5 (7.04%) |
| Male | 34 (12.64%) | 66 (24.54%) | 43 (16.11%) | 45 (16.85%) | 13 (18.31%) | 6 (8.45%) |
| Total | 67 (24.91%) | 124 (46.10%) | 93 (34.84%) | 93 (34.83%) | 21 (29.58%) | 11 (15.49%) |

relationship between child nutritional status with the different household deprivation levels. To assess the magnitude of stunting, wasting, and underweight in different HDS categories, two-way line graphs were generated in STATA. In addition, two-way line graphs were also taken further for disaggregation analysis by girls and boys.

## Results

We have estimated underweight, stunting, and wasting prevalence rates which are 46.1%, 34.83%, and 15.49% respectively in district Rahimyar Khan. Table 2 displays overall underweight, stunting, and wasting rates for Rahimyar Khan disaggregated by the child sex. Underweight and wasting rates for male children were higher as compared to those of female counterparts. In females, stunting rates are higher than those of male children.

Table 3 explicates the prevalence of malnutrition (CIAF) by child age in months. The table shows that 6.35% of malnourished children belong to children of age between 0 to12 month's group. While 8.39% of children are malnourished in the age group between 13 to 24 months, 19.36% of children are malnourished in the age group between 25 to 36 months, 16.45% of children are malnourished in the age group of between 37 to 48 months and 12.68% of children are malnourished in the age group of 49 to 60 months respectively. The first part deals and looks at the distribution of the z-scores with the overall prevalence of malnourishment prevalent in the defined strata groups.

Further, the investigation has been done by comparing the distribution of z-scores with those of the reference population. Fig 1 shows the results of different dimensions of nutritional status in Rahimyar Khan. Fig 1 explains that there are deficits in HAZ and WAZ while only very limited evidence of WHZ (wasting) is present. Fig 2 shows the relationship graphically between different anthropometric indicators. Fig 2 explains that there is no correlation between HAZ and WAZ while there is a minor positive correlation between HAZ and WHZ and also between WAZ and WHZ.

**Table 3. Malnutrition prevalence by age of the child (in months).**

| CIAF | Malnutrition prevalence by age of child (in months) | | | | | |
|---|---|---|---|---|---|---|
| | Not-malnourished | | Malnourished | | Total | |
| | N | % | n | % | n | % |
| 0–12 months | 16 | 5.34 | 19 | 6.35 | 35 | 11.71 |
| 13–24 months | 13 | 4.19 | 26 | 8.39 | 39 | 12.58 |
| 25–36 months | 9 | 2.90 | 60 | 19.36 | 69 | 22.26 |
| 37–48 months | 39 | 12.58 | 51 | 16.45 | 90 | 29.03 |
| 49–60 months | 37 | 11.73 | 40 | 12.68 | 77 | 24.42 |
| | 114 | 36.77 | 196 | 63.23 | 310 | 100 |

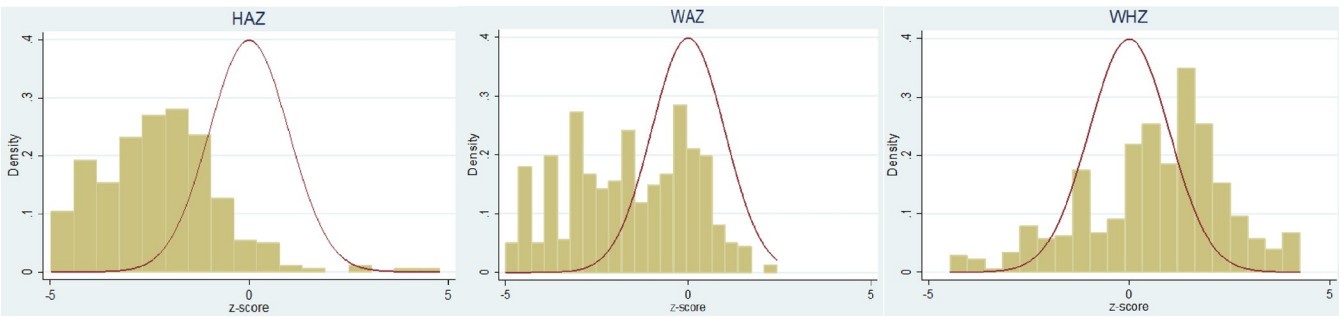

**Fig 1. Distribution of z-scores in district Rahimyar Khan.**

### Household deprivation and pre-school children nutritional status

This section attempts to examine the relationship of anthropometry with different household deprivation levels. Table 4 explains the association among all three indices of child nutrition status with deprivation of household. According to the classification of weight for age, moderate under-weight in all three groups which are HDS-1, HDS-2, and HDS-3 were estimated at 0%, 22.69%, and 2.23% respectively while the severe underweight prevalence in all three groups in pre-school children was estimated at 1.48%, 42.39, and 2.23% respectively. For classification of height for age, moderate stunting prevalence in all three groups was 1.12%, 31.09%, and 2.25% respectively. The prevalence of severe stunting was also the same in all three groups among pre-school children. Furthermore, in weight for height classification, prevalence rates of moderate wasting in all three groups were 1.41%, 25.35%, and 2.82% respectively while severe wasting was estimated at 0%, 14.08%, and 1.41% respectively.

Table 5 illuminates Pearson correlation results for the association between anthropometric indicators with household deprivation status, age, and sex of the child. The p-values show that household deprivation status has a significant association with underweight, stunting, wasting, and CIAF while the age of the child has a significant association with all anthropometric indicators except wasting.

## Discussion and implications

This study constructed HDS based on household socio-economic status and explored the impact of HDS on pre-school child nutritional status regardless of their discrete

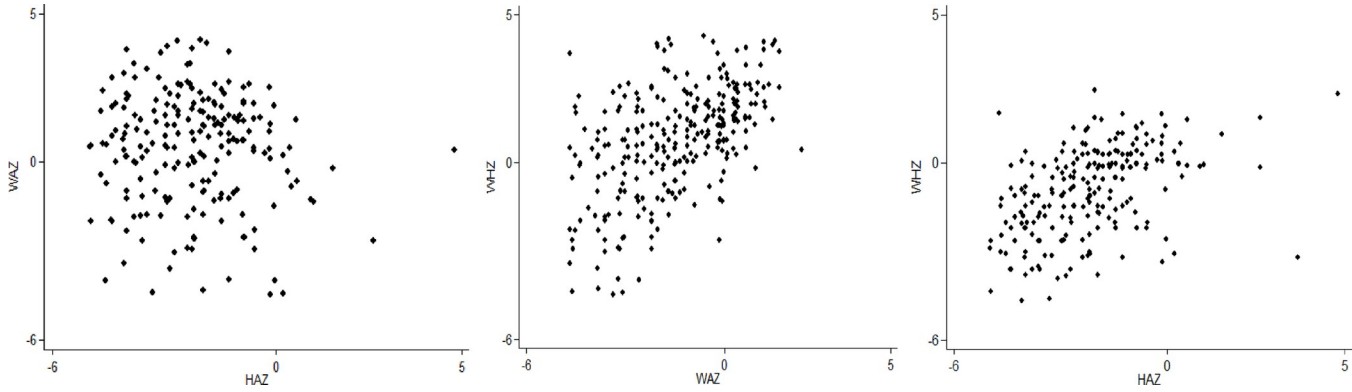

**Fig 2. Correlation between different anthropometric indicators in district Rahimyar Khan.**

**Table 4. Association among child nutrition and HDS.**

| | Nutritional status of child | | | | | | | | |
|---|---|---|---|---|---|---|---|---|---|
| | **Normal** | | **Moderate** | | **Severe** | | **Total** | | |
| **HDS** | **<-1to > -2 z-score** | | **<-2 to > -3 z-score** | | **< -3 z-score** | | | | |
| | **N** | **%** | **N** | **%** | **N** | **%** | **n** | **%** | |
| Weight for age (WAZ) | | | | | | | | | |
| HDS-1 | 4 | 1.48 | 0 | 0 | 4 | 1.48 | 8 | 2.97 | |
| HDS-2 | 66 | 24.54 | 61 | 22.69 | 114 | 42.39 | 241 | 89.59 | |
| HDS-3 | 8 | 2.97 | 6 | 2.23 | 6 | 2.23 | 20 | 7.43 | |
| Total | 78 | 29.00 | 67 | 24.91 | 124 | 46.10 | 269 | 100 | |
| Height for age (HAZ) | | | | | | | | | |
| HDS-1 | 3 | 1.12 | 3 | 1.12 | 3 | 1.12 | 9 | 3.36 | |
| HDS-2 | 72 | 26.97 | 83 | 31.09 | 83 | 31.09 | 238 | 89.15 | |
| HDS-3 | 6 | 2.5 | 7 | 2.5 | 7 | 2.5 | 20 | 7.49 | |
| Total | 81 | 30.34 | 93 | 34.83 | 93 | 34.83 | 267 | 100 | |
| Weight for height (WHZ) | | | | | | | | | |
| HDS-1 | 1 | 1.41 | 1 | 1.41 | 0 | 0 | 2 | 2.82 | |
| HDS-2 | 35 | 49.29 | 18 | 25.35 | 10 | 14.08 | 63 | 88.72 | |
| HDS-3 | 3 | 4.23 | 2 | 2.82 | 1 | 1.41 | 6 | 8.46 | |
| Total | 39 | 54.93 | 21 | 29.58 | 11 | 15.49 | 71 | 100 | |

characteristics. Srinivasan and Mohanty [2] revealed a high influence of household deprivation on preschool children's nutritional status [2, 13]. The results reveal that malnutrition in the Rahimyar Khan District is relatively higher in comparison with the overall national level (S2 Table).

The findings depict that the unavailability of some basic amenities of life at the household level significantly affects the nutritional status of pre-school children. The prevalence of underweight, stunting, and wasting for three groups of household deprivation in Rahimyar Khan as displayed in Fig 3A suggests that the expected tendency of malnutrition is worst for categories of HDS-1 and HDS-2 household socioeconomic deprivation. The study finds that as household shifts from HDS-1 to HDS-2 and then to HDS-3 prevalence rates of stunting and underweight decrease. This is probably due to the improvement in basic amenities of life that takes place. Nonetheless, no substantial impact on the wasting was observed owing to change in the household's socioeconomic status. The main problem of malnutrition in the district is stunting and underweight as the rates of stunting and underweight are higher compared to the rate of wasting. Similarly, evidence was also shown by previous national and provincial surveys in

**Table 5. Association between anthropometric indicators with sex of child, age of child and HDS.**

| Indicators | Underweight | | Stunting | | Wasting | | CIAF | |
|---|---|---|---|---|---|---|---|---|
| | **Chi-sequre value** | **P-value** | **Chi-sequre value** | **P-value** | **Chi-sequre value** | **P-value** | **Chi-sequre value** | **P-value** |
| Sex of Child | 0.0673 | 0.795 | 1.6327 | 0.201 | 0.1793 | 0.672 | 2.1783 | 0.140 |
| Age of Child in Months | 40.5866 | 0.000*** | 23.4537 | 0.000*** | 3.7408 | 0.442 | 23.9913 | 0.000*** |
| HDS | 7.4477 | 0.024*** | 7.2254 | 0.027** | 11.9898 | 0.002*** | 9.6775 | 0.008*** |

Note: *P*-Values are based on chi-square test. Significance level

***p < 0.01

**p < 0.05

*p < 0.1.

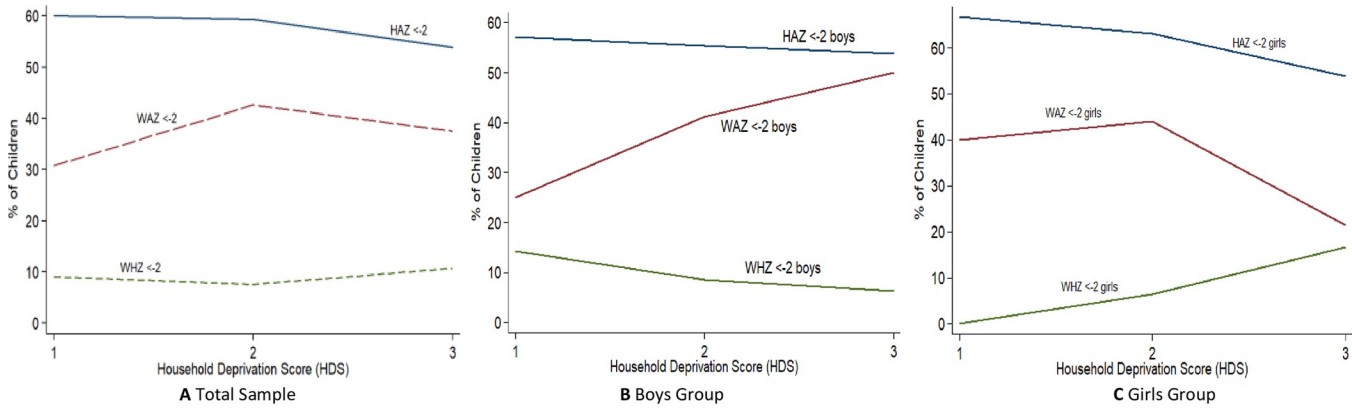

**Fig 3. Prevalence rates of stunting, underweight and wasting for different HDS groups.** A. Total Sample, B. Boys Group, C. Girls Group.

which stunting and underweight rates have remained higher than wasting over time [10, 14–16]. Moreover, Fig 3B and 3C demonstrate the gender-disaggregated analysis. It shows that stunting and wasting rates for male children decrease with the increase in socioeconomic status but the prevalence of underweight remains constant. However, for female children, the incidence of stunting and underweight decreases with the increase in socioeconomic status while wasting remains constant. The disaggregated analysis proved that household deprivation considerably impacted stunting in both genders. Further investigation is needed to inquire about the gender-disaggregated results.

The above-presented visualization of our study is identical to anthropometric measurements of the World Bank [14]. However, the difference is that the World Bank has measured associations of different anthropometrics with wealth index quantiles while this research has correlated anthropometrics with household socioeconomic deprivation index of Srinivasan and Mohanty [2, 13, 14]. The study results are in line with Peruvian Andes [17] which also indicated the strong association of wealth status with stunting showing chances of stunting in the poorest WI quantile were much higher than in the richest quantile.

Malnutrition prevalence rates in children are higher in households belonging to the categories of HDS-1 and HDS-2 as both categories compositely show the most deprived segments of the society in the rural area of Rahimyar Khan. Our results (severe underweight 46%, severe stunting 35%) in rural Rahimyar Khan are similar to previous studies conducted in India that highlight household deprivation impacts child nutrition status in the most deprived segments of society compared to households better off in the basic amenities of life [13]. Using quite a similar index of household assets deprivation, another study in India also found that more than half of truly-poor households have at least one child underweight or stunted in their houses as compared to non-poor counterparts [18]. A similar study in the UK that used a household deprivation index which comprises of six possessions, such as income, employment, health deficiency, and disability, education, skills and training, housing, access to services, has revealed that chances of malnutrition in the most deprived index were greater than those of the least deprived. In addition, patients with medium and high malnutrition risk belong to higher household deprivation areas [19].

Literature indicates that socio-economic deprivation creates inequalities in societies that ultimately lead to malnutrition. Study findings from India [20] showed that wealth status has been one of the key suppliers to socio-economic inequalities in the undernutrition of children over time [20]. Man and Guo [21] showed that child malnutrition was significantly associated with residing in low-level urban communities and low household incomes in China [21]. A

descriptive chi-square analysis of the study found that infant malnutrition was significantly associated with socioeconomic status in the Ecuadorian highlands [22]. Deprivation in basic amenities of life not only influences the nutritional status of children but also affects in case of controlling diseases or restoring the health. For example, a study based in Bangladesh concluded that the odds ratios of rheumatic fever were linked with a low level of income, poor conditions of living, and poor status of nutrition [23]. Using household social deprivation index based on the area of residence, housing, employment of males, low social amenities and car ownership, a study highlighted that malnutrition rates were much higher in most deprived households and these rates were 9.5 percent in most deprived families compared to 6.9 percent in the least deprived families [24]. Also in Kenya, a study observed that deprivation in food and household assets had a strong impact on the nutritional status of children [25].

Most of the research studies in Pakistan also indicated that household poor socio-economic deprivation was a major contributing factor in child malnutrition. A study from rural Swabi, belonging to Khyber Pakhtunkhwa (KPK) province showed household income has a strong association with child malnutrition [26]. Studies from other rural areas of Punjab indicated that low levels of income or poverty caused malnutrition in children [27–29]. Also, for government servants, shopkeepers, and farmers in rural areas of Southern Punjab who usually have low-income, the likelihood of stunting in their children was much higher in comparison to children of landlords [30]. Similarly, researches in nearby areas of rural Sindh showed that children from the poorest households had two times more probability of being wasted and stunted than their counterparts from the wealthier households [31].

The current study highlighted that the rates of underweight (46.1%) and stunting (34.83%) in district Rahimyar Khan are higher compared to the different districts in Punjab. A study in district Multan showed that 18.58% of children were stunted, and 19.54% are underweight [32]. Also, a study in other parts of Punjab depicted that 23.1% of children are underweight, 17.5% are wasted and 28.1% of children are stunted [33]. A study from the federal capital territory Islamabad showed that 29.5% of children were Stunted, 13% wasted, and 35% were underweight [34].

From the discussion, it is evident that there was a direct association between poverty and poor socio-economic deprivation of households with malnutrition. As most of the families in the study belong to the downtrodden class, therefore, social and economic deprivation push the poor into extreme poverty. A study in Rajanpur, the most deprived district of southern Punjab, Pakistan revealed that households having low income had to eat tediously used, expired, and decayed foods. Poor rural households' mothers and children were underprivileged and their food was insecure as they had to put up for sale their highly vigorous food items (milk, honey, purified butter, chicken, and eggs) merely to earn a slight amount of money for other daily necessities [35]. Consequently, the vast rural majority cannot get access to healthy nutritious food and basic amenities of life which leads children towards malnutrition. A report released by Poverty Alleviation Fund and Sustainable Development Policy Institute of Pakistan in 2018 highlighted those eleven districts in three divisions of Southern Punjab have one-fourth of Pakistan's poor population alone and Rahimyar Khan is among these districts and fourth poor district of Punjab having 44% poor population [36, 37]. Around 40% of Pakistanis live below the poverty line, so, poor socioeconomic status is the major cause of malnutrition among the preschool age in Pakistan. If the government improves the social and economic status of the household, they will likely have more resources to provide their children with better food and nutrition along with proper medication in case of any disease.

## Conclusions

This study examined the nutritional status of pre-school children in the rural areas of district Rahimyar Khan and correlated the indicators of the child's nutritional status with household derivational status. Malnutrition was highest in HDS-1 and HDS-2 (the most deprived segments of the population). However, as household deprivation decreased, the rates of stunting (HAZ) and underweight (WAZ) also decreased but wasting remained the same. The measurement of malnutrition through deprivation categories might be more effective than other means. In deprivation, households' access to health and nutritious food becomes hard, and chances of children being underweight increase, and in the long run, children might also become stunted. We urge that deprivation in marginalized districts contributes to malnutrition, which might be eradicated with equal human development opportunities and allocating more budget for underprivileged groups in less developed rural areas. Moreover, the main target of income support programs should also be undernourished households. These steps will not only reduce socio-economic deprivation but also combat undernutrition in the marginalized areas of Pakistan.

## Supporting information

**S1 Table. Distribution of sample size from tehsils to union councils.**
(PDF)

**S2 Table. Comparison of stunting, wasting, underweight prevalance, under-five and infant mortality rates at national, provincial and district levels.** PDHS- 2017–18 (used for national rates), MICS- 2017–18 (used for provincial rates), and Punjab Development Statistics- 2015 (used for district rates).
(PDF)

**S1 Data. Excel file containing the raw data used for analysis in the manuscript.**
(XLSX)

## Acknowledgments

The authors wish to thank all those who kindly volunteered to participate in the study.

## Author Contributions

**Conceptualization:** Farooq Ahmed.

**Data curation:** Muhammad Shahid.

**Formal analysis:** Muhammad Shahid.

**Funding acquisition:** Waqar Ameer.

**Investigation:** Muhammad Shahid.

**Methodology:** Muhammad Shahid, Jing Guo, Saqlain Raza.

**Project administration:** Farooq Ahmed, Waqar Ameer, Jing Guo.

**Resources:** Muhammad Shahid, Saireen Fatima, Madeeha Gohar Qureshi.

**Software:** Muhammad Shahid, Madeeha Gohar Qureshi.

**Supervision:** Farooq Ahmed, Waqar Ameer, Jing Guo.

**Validation:** Farooq Ahmed, Jing Guo, Saireen Fatima, Madeeha Gohar Qureshi.

**Visualization:** Muhammad Shahid.

**Writing – original draft:** Muhammad Shahid, Farooq Ahmed.

**Writing – review & editing:** Farooq Ahmed, Waqar Ameer, Jing Guo, Saqlain Raza, Saireen Fatima, Madeeha Gohar Qureshi.

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
