## [Decision Letter · Decision Letter 0]

22 Oct 2021

PONE-D-21-30029Prevalence of Child Malnutrition and Household Socioeconomic Deprivation: An Empirical Analysis in rural Southern Punjab, PakistanPLOS ONE

Dear Dr. Waqar Ameer,

Thank you for submitting your manuscript to PLOS ONE. After careful consideration, we feel that it has merit but does not fully meet PLOS ONE’s publication criteria as it currently stands. Therefore, we invite you to submit a revised version of the manuscript that addresses the points raised during the review process. Your are requested to submit the revised version of the article keeping the points raised by reviewers 1 and 2 provided with the email.

In addition to comments by reviewers, The following points may also be addressed in the revised version.

1- Why Rahim Yar Khan was chosen for this study? Is there any particular reason?

2- How you can you justify that Only One district (i.e. Rahim Yar Khan) is representative of the entire Sothern Punjab area of Pakistan. It needs to be justified because the title is about southern Punjab. You may revise the title as

"Prevalence of Child Malnutrition and Household Socioeconomic Deprivation: A case study"

3- In Abstract you have written "The results indicate that the prevalence of malnutrition in the Rahimyar Khan is higher than those in other regions of Punjab". Please justify this result as nothing is provided in analysis section that compares situation of Rahim Yar Khan with other areas of Punjab. If you have compared with other area of Punjab than that comparison may added as appendix.

4- Line 45: remove extra space after "[9]"

5- Add some suitable reference to support the text from line 46- line 48: "In Rahim Yar Khan..........their ends."

6- Overall language of the manuscript may be checked for any type of grammatical errors. Please include the following items when submitting your revised manuscript:A rebuttal letter that responds to each point raised by the academic editor and reviewer(s). You should upload this letter as a separate file labeled 'Response to Reviewers'.A marked-up copy of your manuscript that highlights changes made to the original version. You should upload this as a separate file labeled 'Revised Manuscript with Track Changes'.An unmarked version of your revised paper without tracked changes. You should upload this as a separate file labeled 'Manuscript'.

We look forward to receiving your revised manuscript.

Kind regards,

Sajjad Haider Bhatti, Ph.D.

Academic Editor

PLOS ONE

Journal Requirements:

3. PLOS requires an ORCID iD for the corresponding author in Editorial Manager on papers submitted after December 6th, 2016. Please ensure that you have an ORCID iD and that it is validated in Editorial Manager. To do this, go to ‘Update my Information’ (in the upper left-hand corner of the main menu), and click on the Fetch/Validate link next to the ORCID field. This will take you to the ORCID site and allow you to create a new iD or authenticate a pre-existing iD in Editorial Manager. Please see the following video for instructions on linking an ORCID iD to your Editorial Manager account: https://www.youtube.com/watch?v=_xcclfuvtxQ.

4. Please ensure that you include a title page within your main document. You should list all authors and all affiliations as per our author instructions and clearly indicate the corresponding author.

Reviewers' comments:

Reviewer's Responses to Questions

**Comments to the Author**

1. Is the manuscript technically sound, and do the data support the conclusions?

Reviewer #1: Yes

Reviewer #2: Yes

2. Has the statistical analysis been performed appropriately and rigorously? 

Reviewer #1: Yes

Reviewer #2: Yes

3. Have the authors made all data underlying the findings in their manuscript fully available?

Reviewer #1: Yes

Reviewer #2: No

4. Is the manuscript presented in an intelligible fashion and written in standard English?

Reviewer #1: Yes

Reviewer #2: Yes

5. Review Comments to the Author

Reviewer #1: The paper uses reasonable empirical design and analysis method to prove the correlation between household social-economic status and children's malnutrition condition. Three commonly-used and widely-acknowledged measurements of malnutrition are used in the paper, which are standardiezed z-score of weight-for-age (WAZ), weight-for-height(WHZ) and height-for-age (HAZ). The data used in the article is collected via rigorous stratified sampling method in the most deprived area in rural Pakistan, which matches the academic question the paper aims to address. Therefore, I recommend this article for acceptance. The research could be extended in the following areas.

1. The empirical analysis could be exteneded to childeren within other aging range. Malnutrition occurs at all age levels of teenagers of the developing world.

2. The channel and mechanism of the impact of socialeconomic gradients on the children's health outcomes could be another question of interest.

3. More heterogeneity could be added to the research if possible in the future. For example, results of children of different birth orders could be tested. Other measurements of socialeconomic status like projected lifetime income, or consumption, or occupation could be also used in the empirical study.

Reviewer #2: The article titled “Prevalence of Child Malnutrition and Household Socioeconomic Deprivation: An Empirical Analysis in rural Southern Punjab, Pakistan” has studied the hypothesis that household poverty is responsible for the child malnutrition.

The study is quite interesting in the purview of deprived rural area from Pakistan. It is the source to increase the literature from African countries to South Asia. In my view, the article is well-written and interesting especially for low- and middle-income counties context. After the following corrections/reviews, the article can add value to the journal and for the researchers in South Asian countries.

The author is using the word ‘under five years’ age of children unnecessarily many a times in the article. However, it is repeated multiple times in the Methods section, too (Page 3, line 75, 80; Page 4, line 87, 102-103). By studying the sample characteristics, it is understood. Author should eliminate it wherever it irritates the reader (e.g. Page 6, line 147).

Page 4, line 110: null hypothesis assumes that a child comes from a healthy population. The hypothesis does not clearly show the relation with study objectives, research question, or study hypothesis. Author needs to clearly write this hypothesis according to the study objectives.

Page 15, In Table S1, the sum of last column numbers is not correct.

In the analysis, p-values can add some stronger evidence of significance, if added in Tables 2, 3, and 4.

What is the importance of results disaggregated by sex. What are the relevant recommendations in the article?

6. PLOS authors have the option to publish the peer review history of their article (what does this mean?). If published, this will include your full peer review and any attached files.

Reviewer #1: No

Reviewer #2: No

---

## [Author Response · Author response to Decision Letter 0]

31 Dec 2021

Dear Editor

PLOS ONE

We are thankful for your comments and inputs.

We have revised the entire paper very carefully and improved its write-up and material. Data are shared in accordance with participant consent and all applicable local laws. Data availability staement has been incorporated in the manuscript. All the relevant data is in the manuscript and its supporting information file. We have uploaded the excel file and deidentified participants’ information. In our opinion there is no such data in the file that can identify our participants. The Authors' information and their contribution are given below. 

Comments by the academic editor

1- Why Rahim Yar Khan was chosen for this study? Is there any particular reason?

Response by authors: Thanks for your valuable comment. The reason behind choosing district Rahimyar khan as a case study was that Rahimyar Khan is the fourth poor district of Punjab in which nearly 44% population is poor [36, 37], 78% of people live in the rural areas [9] and the prevalence of malnutrition and child mortalities is the third-highest among the thirty-six (36) districts in Punjab [10]. About 93% of the households belonged most deprived segment of the society in terms of basic amenities of life. For this reason, the district was taken to verify the link between child nutritional status with the different household deprivation levels by arguing whether household deprivation status is contributing to child malnutrition. (See line 48-50 & 271-74) 

2- How can you justify that Only One district (i.e., Rahim Yar Khan) is representative of the entire Sothern Punjab area of Pakistan. It needs to be justified because the title is about southern Punjab. You may revise the title as "Prevalence of Child Malnutrition and Household Socioeconomic Deprivation: A case study"

Response by authors: Thanks for your valuable comment. Now, the title is revised in the final draft which is “Prevalence of Child Malnutrition and Household Socioeconomic Deprivation: A Case Study of Marginalized District in Punjab, Pakistan”. One district could not be representative of the entire region, so in the final revised draft, we have corrected this narrative and focused on the district only. 

3- In the Abstract you have written "The results indicate that the prevalence of malnutrition in the Rahimyar Khan is higher than those in other regions of Punjab". Please justify this result as nothing is provided in analysis section that compares situation of Rahim Yar Khan with other areas of Punjab. If you have compared with other area of Punjab than that comparison may added as appendix.

Response by authors: Thank you for this valuable comment. The statement in the abstract section was based upon Multiple Indicator Cluster Survey-2014 but not on our results. The statement was mistakenly written. We corrected our mistake in the abstract now.

However, as you pointed out our results in district Rahim Yar Khan are not compared with other districts of Punjab, we also filled that gap and discussed our results with different regions of Punjab in the discussion section [See line 250-261]. Furthermore, a comparison of Malnutrition prevalence and child mortalities of district Rahimyar Khan with average rates in both Pakistan and Punjab is given in Table-S2 in the revised draft.

4- Line 45: remove extra space after "[9]"

Response by authors: We are thankful for your valuable comment. We have removed all the extra spaces in the revised final draft including “this one you mentioned.”

5- Add some suitable reference to support the text from line 46- line 48: "In Rahim Yar Khan..........their ends."

Response by authors: Thanks for your valuable comment. We added the reference in the introduction section (see reference 10 in the revised manuscript; also this reference is given below)

“Punjab Bureau of Statistics. Punjab 2014 Multiple Indicator Cluster Survey Key Findings Report. Lahore (2014).”

6- Overall language of the manuscript may be checked for any type of grammatical errors.

Response by authors: Thanks for your valuable comment. We have now very carefully revised the final draft multiple times to ensure that no language and grammatical errors are there. 

Comments by Reviewer #1

1- The empirical analysis could be extended to children within other aging range. Malnutrition occurs at all age levels of teenagers of the developing world.

Response by authors: Thanks for your valuable comment. We have only assessed the malnutrition status of only 5 years children. There is consensus in nutritional research, children under five years of age, being dependent on parents, need more care, and are more prone to diseases and infections, so, the chances of a child being malnourished are higher in this life span. This was the main reason our study also has assessed the nutritional status of under-five children only. 

2- The channel and mechanism of the impact of socioeconomic gradients on the children's health outcomes could be another question of interest.

Response by authors: Thanks for your valuable comment. We only focused on the close association between household deprivation with malnutrition status by hypothesizing that the relationship of Household Deprivation Status (HDS) would be strong with child malnutrition in district Rahimyar Khan. The HDS does not directly measure household economic conditions like total expenditure, per-capita income, or living standard index, rather it measures households on the three dimensions that are deprived. Household deprivation explains the socioeconomic status of a household. It is an alternative index for income status or wealth status or living standard index. 

3- More heterogeneity could be added to the research, if possible, in the future. For example, results of children of different birth orders could be tested. Other measurements of socioeconomic status like projected lifetime income, or consumption, or occupation could be also used in the empirical study.

Response by authors: Thanks for your valuable comment. We hypothesized that the relationship of Household Deprivation Status (HDS) would be strong with child malnutrition. The comment somehow indicates multivariate analysis for malnutrition. There is a lot of literature on it. The existing literature in Pakistan only guides us that wealth status or poor socio-economic status is one of the significant determinants of malnutrition. But no one has seen the in-depth association between malnutrition and poor socio-economic status (household deprivation) in Pakistan which is an objective of study.

Comments by Reviewer #2

1- The author is using the word ‘under five years’ age of children unnecessarily many a times in the article. However, it is repeated multiple times in the Methods section, too (Page 3, line 75, 80; Page 4, line 87, 102-103).

 Response by authors: Thanks for your valuable comment. We eliminated the repetitive word and replaced it with suitable alternatives to remove redundancy.

2- By studying the sample characteristics, it is understood. Author should eliminate it wherever it irritates the reader (e.g., Page 6, line 147). 

Response by authors: Thanks for your valuable comment. We eliminated it in the final draft.

3- Page 4, line 110: null hypothesis assumes that a child comes from a healthy population. The hypothesis does not clearly show the relation with study objectives, research question, or study hypothesis. Author needs to clearly write this hypothesis according to the study objectives.

Response by authors: Thanks for your valuable comment. We have corrected it and now the null hypothesis assumes that a child under study is not malnourished. 

4- Page 15, In Table S1, the sum of last column numbers is not correct.

Response by authors: Thanks for your appreciated comment. It is corrected in table S1 now, the household sample of tehsil Sadiqabad is 92 which was previously written mistakenly equal to 115.

5- In the analysis, p-values can add some stronger evidence of significance, if added in Tables 2, 3, and 4.

Response by authors: Thanks for your valuable comment. We have already incorporated p-values in Table 5 that produce the same meaning as if we calculate p-values for Tables 2, 3, and 4. If we calculate p-values for these tables, we think that the results will be unnecessarily doubled and can be misunderstood by the reader.

6- What is the importance of results disaggregated by sex. What are the relevant recommendations in the article?

Response by authors: Thanks for your valuable comment. The theme of this paper is to check the tendency of malnutrition prevalence according to the household deprivation status by hypothesizing that the relationship of Household Deprivation Status (HDS) would be strong with child malnutrition in district Rahimyar Khan. Our primary objective is achieved when we check it with overall children. However, we just estimated the behavior of malnutrition prevalence by gender to see if gender is significant or not in deprived households but its detailed analysis might be covered in further research as it was not our prime objective.

---

## [Editor Report · Decision Letter 1]

20 Jan 2022

Prevalence of Child Malnutrition and Household Socioeconomic Deprivation: A Case Study of Marginalized District in Punjab, Pakistan

PONE-D-21-30029R1

Dear Dr. Ameer,

We’re pleased to inform you that your manuscript has been judged scientifically suitable for publication and will be formally accepted for publication once it meets all outstanding technical requirements.

Kind regards,

Sajjad Haider Bhatti, Ph.D.

Academic Editor

PLOS ONE

Additional Editor Comments (optional):

In my opinion, the authors have addressed all concerns raised by me (as academic editor) and reviewers.

So, now I am in a position to recommend acceptance of the manuscript for publication in PLOS ONE.

Thanks and Regards
---

## [Editor Report · Acceptance letter]

1 Mar 2022

PONE-D-21-30029R1 

Prevalence of child malnutrition and household socioeconomic deprivation: A case study of marginalized district in Punjab, Pakistan 

Dear Dr. Ameer:

I'm pleased to inform you that your manuscript has been deemed suitable for publication in PLOS ONE. Congratulations! Your manuscript is now with our production department. 

Kind regards, 

on behalf of

Dr. Sajjad Haider Bhatti 

Academic Editor

PLOS ONE